# Macro, Trace and Toxic Element Composition in Liver and Meat of Broiler Chicken Associated with Cecal Microbiome Community

**DOI:** 10.3390/biology13120975

**Published:** 2024-11-26

**Authors:** Dmitry Deryabin, Dianna Kosyan, Ludmila Vlasenko, Christina Lazebnik, Alexander Zatevalov, Ilshat Karimov, Galimzhan Duskaev

**Affiliations:** 1Laboratory of Breeding and Genetic Research in Animal Husbandry, Federal Research Centre of Biological Systems and Agrotechnologies, 460000 Orenburg, Russia; kosyan.diana@mail.ru (D.K.); lv.efremova@yandex.ru (L.V.); christinakondrashova94@yandex.ru (C.L.); zatevalov@mail.ru (A.Z.); ifkarimov@yandex.ru (I.K.); gduskaev@mail.ru (G.D.); 2Laboratory for Diagnostics and Prevention of Infectious Diseases, G.N. Gabrichevsky Research Institute for Epidemiology and Microbiology, 125212 Moscow, Russia; 3Department of Epidemiology and Infectious Diseases, Orenburg State Medical University of the Ministry of Health of Russia, 460014 Orenburg, Russia

**Keywords:** essential elements, toxic elements, gut microbiota, microbiome pattern, broiler chicken

## Abstract

Microbiota in the chicken gastrointestinal tract affect the elemental composition of the chicken’s body through macro, trace and toxic element availability and absorption, as well as competition with the host biogenic elements. The current study presents a meta-analysis of the detailed relationship between the composition of 25 essential and toxic elements in chicken tissues examined by ICP-MS and the gut microbial community analyzed using NGS techniques. The examination of chicken liver and meat revealed typical elemental compositions, called the “elementomes” (α, β, γ), which were generally stratified, not continuous. An in-depth broiler chicken cecal microbiome analysis revealed nine bacterial phyla, divided into 16 classes, 26 orders, represented by 62 families and, at the lowest taxonomic level, including 130 genera. Following characterization of the microbiomes’ structure, there were two distinct enterotypes indicated, designated “microbiome patterns”. This article provides the first description of the multiple strong associations between the gut microbial community and the body elemental composition of broiler chickens. This insight proposes a novel strategy to improve deficiency or excess of certain elements in the host by gut microbiome modulation, which needs to be verified with further in vivo experiments.

## 1. Introduction

Chicken meat, liver and other by-products present several advantages, namely a delicious taste, high level of safety, and relatively affordable price [1]. For these reasons, broiler chickens are currently one of the main sources of animal protein [2], with global production growing dynamically from 83,267 million metric tons in 2012 to 103,418 million metric tons by 2023 [3].

Poultry products, in addition to protein, also contain a wide range of macro elements (Na, K, Mg, Ca, P), as well as some trace elements such as chromium (Cr), manganese (Mn), iron (Fe), cobalt (Co), copper (Cu) and zinc (Zn) [4], which are essential to maintain correct metabolic activity and balance body functions and immunity. On the other hand, chicken tissues can accumulate a number of toxic elements such as lead (Pb), arsenic (As) and cadmium (Cd), which when transmitted through the food chain pose a potential hazard to human health [5]. Thus, the determination of macro, trace and toxic elements in poultry products is becoming an urgent experimental and applied scientific problem that is intensively studied worldwide [6,7,8].

Several factors that influence the elemental content in the chicken’s body have been described. The first significant factor is the chicken’s genome, which encodes key pathways associated with feed efficiency and related to the transportation, deposition and excretion of macro, trace and toxic elements [9]. In turn, the second significant factor is the type of diet, which can have different elemental contents [10], varying in the level of proteins, fats and carbohydrates [11] or supplemented with enzymes and antibiotics [12].

A novel aspect of the problem under discussion is the role of microbiota in the chicken gastrointestinal tract, which affects the elemental composition of the host body through the availability of macro, trace and toxic elements by controlling the pH and rH in the gut lumen [13], modulating the elements’ absorption by gut villi [14], and promoting competition between micro- and macroorganisms for biogenic elements [15].

In the current context, it is significant that various dietary additives containing live probiotic strains, antimicrobials or small molecules that affect density-dependent communication in bacteria can change the gut microbiota composition and thereby indirectly influence mineral, protein and fat metabolism, inflammatory status and the chicken’s growth performance [16,17]. Moreover, the chicken’s gut microbiome also depends on the host genetics [18], which conceptualizes a multifactorial hierarchical regulatory cascade of the elemental composition of the chicken body (Figure 1).

Several bacterial taxa with proven influence on the accumulation of some elements in the host body have been reported [19], while similar effects of multicomponent gut bacterial communities named “enterotypes” [20] or “microbiome patterns” [21] have not yet been studied sufficiently. A limited number of articles have linked microbial enterotypes to macro, trace and toxic elements in humans [22], while studies examining the relationship between microbiome patterns and the elemental composition of broiler chickens are not available.

In our previously published studies, we have reported several experimental series focused on the elemental composition of chicken’s liver and meat [23,24,25,26], as well as the microbial community of broiler cecum assessed using NGS metagenomics techniques [27,28,29,30,31,32,33,34,35]. The variability of numerous determined parameters was shown; however, the associations between the cecal microbiome and the macro, trace and toxic element content in the chicken’s body has not been analyzed. This study aimed to perform a meta-analysis of summarized data to explore the detailed relationship between the composition of 25 essential and toxic elements in chicken’s tissues and observed the gut microbial community.

## 2. Materials and Methods

### 2.1. Ethics Statement

The study protocols were approved by the Animal Ethics Committee of the Federal Research Centre of Biological Systems and Agro-Technologies of the Russian Academy of Sciences, following Article 2(d) of the European Convention for the Protection of Vertebrate Animals used for Experimental and other Scientific Purposes (18 March 1986) and the principles of good laboratory practice.

### 2.2. Data Collection: Animals and Feed

In most experimental series [26,27,28,29,30,31,33], the Arbor Acres cross (Aviagen LLC, Tula, Russia) was used. In some experimental series [34,35], the Smena-8 cross (Center for Genetics and Selection “Smena”—branch of Federal Scientific Center “All-Russian Research and Technological Institute of Poultry” of Russian Academy of Sciences, Moscow region, Russia) was used.

After the post-hatch period, 7-day-old male broiler chickens were randomly divided into control and experimental groups of 15–30 poultries each, marked with plastic foot tags and kept in specialized cages with an area of 4050 cm^2^ (90 × 45 × 45 cm, 3 tiers) with a mesh floor. Temperature and relative humidity regimes corresponded to the requirements for growing broilers, and the photoperiod program (23 h continuous light 30 L: 7 D, 20 h continuous light 10 L: 8–20 D, and 18 h continuous light 8 L: 21–45 D) complied with the European Council Directive (2007/43/EU), which regulates the protection of chickens kept for meat production.

According to Arbor Acres Broiler Nutrition Specification (2022), the chickens were kept on a two-phase basal diet throughout the experimental period: the grower diet from days 8–28 and the finishing diet from days 29–42. The granulated compound feed was represented by pellets of 2.5–3.0 mm. The content of macro, trace and toxic elements in the daily basic diet was described by Lebedev et al. [36]. The control group in each series received only the basic diet, which did not contain any antibiotics, probiotics or phytogenic compounds, while the experimental groups were fed a diet supplemented with Oak Bark extract [34,35], *Bacillus*-containing probiotic additives [27], the antibiotic chlortetracycline [29,30,35], or chemically synthesized analogues of the following plant-derived molecules: gamma-lactone, coumarin derivative, vanillin/vanylic acid, resorcinol derivative, or quercetin [26,27,28,29,30,31,33]. A detailed description of the additives and their use in poultry feeding is presented in the additional table placed in Appendix A. It is important for the current content to specify that these supplements affected the cecal microbiome composition but did not contain macro, trace or toxic element additives.

At the end of the experiments, randomly selected chickens from each group were humanely euthanized and aseptically dissected. Their liver (typically the caudal part of the right lobe), thigh (central part of the *pectoralis major* muscle) and breast (central part of the *iliotibialis* muscle) were carefully excised and samples were placed in plastic tubes. The cecal contents were massaged into sterile cryogenic tubes, snap frozen in dry ice and stored at −80 °C in an 803CV freezer (Thermo Fisher Scientific, Waltham, MA, USA).

### 2.3. Determination of Elements: Samples Preparation and Analytical Procedure

For the determination of elements, 1 g of each tissue sample (liver, thigh, or breast) was digested with 5 mL of concentrated HNO_3_ (Sigma-Aldrich, St. Louis, MO, USA) and combusted using a Multiwave 3000 device (Anton Paar, Graz, Austria) in the following mode: raising to 200 °C for 5 min, holding at 200 °C for 5 min and cooling to 45 °C. Dissolved samples were transferred into 15 mL test tubes, the volume was brought to 10 mL with distilled deionized water (18 MOM cm^−1^) (Merck KGaA, Darmstadt, Germany), the tubes were closed and the samples mixed. Using an automatic dispenser, aliquots of 1 mL were taken, adjusted to 10 mL with 0.5% nitric acid, and submitted for analysis.

Quantitative determination of macro elements (Ca, K, Mg, Na, P), essential trace elements (Co, Cr, Cu, Fe, I, Li, Mn, Se, Si, Sr, V, Zn) and toxic trace elements (As, Al, B, Cd, Hg, Ni, Pb, Sn) was carried out using a NexION 300 D quadrupole inductively coupled plasma optical emission spectrometer or an Optima-2000 DV atomic emission spectrometer (Perkin Elmer, Waltham, MA, USA) according to the manufacturer’s recommendations. The determined element content in the liver and meat of broiler chickens was presented as ‰ for macro elements or ppm for trace and toxic elements.

To calibrate the equipment, Universal Data Acquisition Standards Kits (Perkin Elmer, Waltham, MA, USA) with different element contents were used. All preparative and analytical procedures were carried out in accordance with quality management standards, which ensured good reproducibility within individual experimental series and data comparability between different experimental series.

The experimental data were compared with the maximum permissible levels of essential and toxic elements recommended by the Food and Agriculture Organisation (FAO)/World Health organization (WHO) summarized by Korish and Attia (2020) https://openknowledge.fao.org/server/api/core/bitstreams/6e2d2772-5976-4671-9e2a-0b2ad87cb646/content (accessed on 30 October 2024).

### 2.4. Microbial Community: DNA Extraction, Sequencing, and Data Processing

Total DNA extraction from cecal samples in two experimental series [27,34] was performed according to the phenol–chloroform protocol, while the Fast DNA SPIN Kit for Feces (MP Biomedicals Inc., Irvine, CA, USA) and QIAamp Fast DNA Stool Mini Kit (Qiagen GmbH, Hilden, Germany) was used in four [26,29,30,31] and three [28,33,35] series, respectively. The extracted DNA quality was evaluated by 1% agarose gel electrophoresis, and DNA concentration was assessed using the dsDNA high-sensitivity assay kit for a Qubit 4 fluorometer (Life Technologies, Carlsbad, CA, USA) according to the manufacturer’s recommendation.

Illumina metagenome sequencing technology focused on the V3-V4 region of the 16S rRNA gene was used to characterize the cecal microbiome compositions. The 16S Metagenomic Sequencing Library Preparation two-stage protocol (Part #15044223, Rev. B) was implemented, and the forward S-D-Bact-0341-b-S-17 (CCTACGGGNGGCWGCAG) and the reverse S-D-Bact-0785-a-A-21 (GACTACHVGGGTATCTAATCC) primers were used for amplification of the targeted V-regions. The prepared DNA was purified by solid-phase reversible immobilization on Agencourt AM Pure XP beads (Beckman Coulter, Brea, CA, USA), after which the quality of the DNA libraries was confirmed using the QIAxcel Advanced system (Qiagen GmbH, Hilden, Germany). High-throughput paired-end 2 × 250 bp sequencing with the V.2 reagent kit (Illumina Inc., San Diego, CA, USA) was performed on the MiSeq platform.

Raw sequencing data processing via reads merging into contigs, filtering by length (at least 420 bp) and quality (maxee 1.0), chimera deletion, dereplication, and clustering into separate operational taxonomic units (OTUs) was performed using the USEARCH V.10.0.240 software. The OTUs were assigned by the RDP classifier [37] against the SILVA 16S rRNA database (http://www.arb-silva.de/) accessed on 1 June 2024. The final OTUs’ taxonomic affiliation at the phylum, class, order, family and genus level was determined according to the Genome taxonomy database (GTDB) (https://gtdb.ecogenomic.org/) accessed on 1 June 2024 and the NCBI taxonomy browser [38].

### 2.5. Statistical Analysis

A total of 36 groups of broiler chickens (7 control and 29 experimental), representing 9 experimental series, were included in the study. A total of 25 elements, including 5 macro elements (Na, K, Mg, Ca, P), 12 essential trace elements (Li, Sr, Si, Se, I, V, Cr, Mn, Fe, Co, Cu, Zn) and 8 toxic elements (B, Al, Sn, Pb, As, Ni, Cd, Hg), were determined in liver, thigh and breast samples. Complete data sets were collected for 29 broiler chicken groups, while data for 7 groups were incomplete for Sr, Al, Sn, Pb, Cd and Hg, among which As was not determined in 4 groups. Complete microbiome data were collected for all control and experimental groups. Meta-analysis was performed using pre-calculated data for each group as an analyzed unit.

The first round of meta-analysis consisted of descriptive statistics that quantify differences in element content and microbiome composition between groups. Because the Shapiro–Wilk test rejected the normal distribution hypothesis of the original data with 95% confidence, nonparametric procedures for downstream statistical analyses were justified. Data for the general population and selected subgroups were presented as median values [Q1–Q3], and the difference between data samples was assessed using the Mann–Whitney U test, where *p* values < 0.001 were considered significant.

The second round of meta-analysis, aimed at assessing of macro, trace and toxic element compositions in various biosamples as well as cecal microbiome communities, was conducted using a multivariate statistical method called principal component analysis (PCA). The groups’ distribution by element compositions or microbiome communities was visualized via F1 × F2 factor plane. The influence of bacterial phyla on the macro and trace element accumulation in various biosamples was represented by the R2 coefficient of determination.

Statistics and visualization were performed using the Statistica 13.0 software package (TIBCO Software Inc., Palo Alto, CA, USA), and raw and derived data were stored using Microsoft Excel 2019 (Microsoft Corporation, Redmond, Washington, DC, USA). The associations between the element contentment in chicken biosamples and bacterial genera in the chicken cecal microbiomes were visualized via a “heat map” chart using the BioVinci data package.

## 3. Results

### 3.1. Macro, Trace and Toxic Element Content in the Liver and Meat of Broiler Chickens

The determined concentrations of 25 elements in the examined broiler chicken biosamples are presented as median values [Q1–Q3] in Table 1.

When liver and meat samples were compared in terms of macro element content, liver samples showed statistically significant (*p* < 0.001) high concentrations of Na, Ca and P, while thigh and breast samples were characterized by increased K and Mg content compared to the liver (*p* < 0.001), but for all five macro elements, these did not differ from each other.

Trace elements analysis revealed no differences between liver and meat, as well as between thigh and breast samples in the content of five elements: Li, Sr, Si, I and Cr. In turn, the most significant differences were found for Mn, of which the concentration in the liver was 15.84-fold higher than in the thigh and 16.02-fold higher than in the breast (*p* = 3.0 × 10^−13^), as well as for Fe, of which the concentration was 14.79-fold and 11.29-fold higher (*p* = 3.0 × 10^−13^), respectively. Additionally, the Se, V, Co, Cu and Zn contents in the liver were also significantly increased compared to meat (*p* < 0.001), while those in the thigh and breast were not statistically different. When compared with the FAO/WHO maximum permissible levels of essential trace elements in poultry products [4], Cr, Mn, Cu and Zn were assessed as being excessive in liver samples, while half of the meat samples had excess Zn levels, about a third showed an excess of Mn and Cu, and less than a quarter contained increased Se concentrations.

The values of most examined toxic elements (B, Sn, Pb, As, Ni, Hg) were the same in all liver and meat samples. The only differences reported were increased Al levels in the breast vs. liver (*p* = 6.0 × 10^−5^) and decreased Cd concentration in the thigh and breast vs. the liver (*p* = 1.1 × 10^−11^ and *p* = 6.8 × 10^−5^, respectively). Comparison with the permissible limits set by FAO/WHO indicated only a sporadic increase in the maximum permissible level of Pb in 3.45% and Ni in 2.78% of breast samples.

### 3.2. Elemental Compositions of Broiler Chicken Biosamples

In order to analyze the inter-elements’ tendency of accumulation in tissues of broiler chickens, correlation coefficients were calculated for liver, thigh and breast samples. The most significant values (r ≥ 0.9; very strong correlations) are indicated in Figure 2A–C.

In the liver, 182 statistically significant (*p* < 0.001) associations were identified, including 136 positive and 46 negative correlations. Numerous very strong correlations (43 with a value of r ≥ 0.9) linked several of the determined macro (Na, K, Mg, P) and trace (Sr, Se, Mn, Co, Cu, Zn) elements, some of which showed associations with single toxic elements (B, Pb, Ni). In turn, Si and Cr were strongly positively correlated with each other, but were negatively associated with most of the elements listed above when combined with V. A similar correlation network was found in the thigh, where 94 positive and 22 negative correlation coefficients were recorded with a *p*-value < 0.001, including 24 very strong correlations (r ≥ 0.9) between macro elements, several trace elements (Mn, Cu, Zn) and certain toxic elements (B, Al, Cd). When a correlation analysis of breast samples was performed, the number of statistically significant (*p* < 0.001) correlations decreased to 75 (72 positive and 3 negative); among them, the strongest associations were shown for macro elements (Na, K, Mg, P) and Cd.

Subsequent principal component analysis transformed the multiple identified correlations into a new coordinate system that allowed us to identify the factors (components) capturing the most characteristic variations in the complete data sets. As shown in the scree plots (Figure 2D–F), most scalar projections of the data come to lie on the first coordinate (called the F1 factor or first principal component), whose correlation matrix eigenvalues were 16.02 for the liver samples, 14.07 for thigh samples and 12.20 for breast samples (total variance—64.11%, 56.26% and 48.81%, respectively). The F1 component structure typically integrated high positive factor loadings (>0.7) for all of the examined macro elements, several trace elements (Sr, Mn, Cu; optionally Se, Fe, Zn) and some toxic elements (B; optionally Al, Cd), while the factor loadings for Si, V and Cr had strictly negative values (Figure 2G–I).

The calculation of factor scores made it possible to link the analyzed groups with the F1 factor directly or inversely. In turn, the projection of the macro, trace and toxic element compositions of broiler chicken biosamples onto the F1 × F2 factor plane showed that elemental variation is generally stratified, not continuous. For example, Figure 3 shows three robust clusters of chicken liver samples, which differ significantly from each other in elemental content, and according to Peñuelas et al. (2019) [39] these are called “elementomes”.

The α-elementome, which has positive F1L factor scores and was found in eight chicken groups, was characterized by a high content of macro elements (Na, K, Mg, Ca, P), majority trace elements (Sr, Se, Mn, Fe, Co, Cu, Zn) and some toxic elements (B, Pb, Ni, Cd), with a deficiency of Si, V and Cr. The alternative β-elementome, integrating data sets from 18 groups, on the contrary, had negative F1 L factor scores and was characterized by high accumulation of Si, V and Cr with a decreased content of most of the macro, trace and toxic elements mentioned above. In turn, the γ-elementome was relatively indifferent to the F1 L factor, but had a positive F2 L factor score, which indicates an increase in the content of Li and a number of toxic elements (Al, As and Hg). Similar elementomes were also found in the analysis of thigh and breast samples, with some trace elements and toxic elements optionally included or excluded.

### 3.3. Bacterial Community in Chicken Cecal Microbiomes

Nine bacterial phyla were found in the chicken cecal microbiomes, defined as monophyletic lineages of bacteria. Their 16S rRNA genes have a pairwise sequence identity of approximately 75% or less with those of other bacterial phyla. At this highest taxonomic level, the prevalent phyla in all groups were *Bacillota* and *Bacteroidota*, which occurred in 100% of cecal samples and showed relative abundances in the bacterial community of 71.45% [49.24–80.64] and 26.81% [14.27–49.51], respectively (Figure 4). Resident members of the core chicken cecal microbiomes also included the phyla *Actinomycetota*, *Cyanobacteriota*, *Pseudomonadota* and *Thermodesulfobacteriota*, which were present in at least two-thirds of the analyzed samples but showed low abundance (less than 1%). Three more phyla (*Chloroflexota*, *Mycoplasmatota* and *Verrucomicrobiota*) were found sporadically. In turn, about 0.23% [0.06–0.44] of operational taxonomic units could not be correctly characterized in terms of phylum definition and were therefore assessed as unclassified bacteria.

The following taxonomic classification divided these phyla into 16 classes, 26 orders, represented by 62 families and, at the lowest taxonomic level, including 130 bacterial genera.

A total of 59 bacterial genera and taxa, finally classified at the family level, were found in at least 50% of the samples analyzed, showing them as resident members of the core cecal microbiome. The top 20 genera with the highest occurrence and relative abundance are presented in Table 2.

This list includes the genera *Bacteroides* and *Alistipes*, belonging to the phylum *Bacteroidota*, as well as *Butyricicoccus*, *Faecalibacterium*, *Lactobacillus*, *Ruminococcus*, *Christensenella* and some other genera belonging to the phylum *Bacillota*.

### 3.4. Cecal Microbiome Communities (Microbiome Patterns)

Co-occurrence of different bacterial taxa showing symbiotic or oppositional relationships between cecum inhabitants was revealed using correlation analyses, and typical microbiome communities (patterns) were determined based on principal component analyses.

The correlation coefficient calculation revealed six significant (*p* < 0.001) associations between five bacterial phyla included in the core cecal microbiome (Figure 5A). On the one hand, positive correlations were found between the phyla *Bacillota* and *Cyanobacteriota* (r = 0.57, *p* = 3.0 × 10^−4^), as well as between *Actinomycetota* and *Thermodesulfobacteriota* (r = 0.88, *p* = 8.1 × 10^−13^), while these symbiotic pairs were not directly associated with each other. On the other hand, multiple negative correlations were found between the phylum *Bacteroidota* and *Actinomycetota* (r = −0.54, *p* = 6.2 × 10^−4^), *Bacillota* (r = −0.99, *p* = 3.8 × 10^−30^), *Cyanobacteriota* (r = −0.54, *p* = 5.9 × 10^−4^) and *Thermodesulfobacteriota* (r = −0.58, *p* = 2.3 × 10^−4^), which indicated oppositional relationships between these taxa.

These associations were supported by correlation networks at low taxonomic levels, where 583 positive and 24 negative significant (*p* < 0.001) correlation coefficients were identified between various bacterial genera.

Processing the data using principal component analysis, the first weight factor F1 was described, whose correlation matrix eigenvalues were 3.34 and the total variance was 33.41% (Figure 5B). The F1 structure included significant unrotated factor loadings (>0.07) for the four bacterial phyla (Figure 5C). The highest positive and negative factor loadings were found for *Bacillota* (+0.91) and *Bacteroidota* (−0.94), which confirms the oppositional relationships between these taxa. The F1 structure also included the phyla *Actinomycetota* and *Thermodesulfobacteriota* (factor loadings 0.71 and 0.73, respectively) complemented by the phylum *Cyanobacteriota* due to its strong positive correlation with the phylum *Bacillota* (see above).

Thus, principal component analysis confirmed two previously reported [21] microbiome patterns: F1(−) enriched in *Bacteroidota* and F1(+) dominated by *Bacillota* and coupled with members of the phyla *Actinomycetota*, *Cyanobacteriota* and *Thermodesulfobacteriota*, therefore designated as *Bacillota* + ACT compositions. The following calculation of factor scores associated each chicken group with alternative F1(−) or F1(+) patterns, and their projection onto the F1_MB_ × F2_MB_ factor plane showed a clear distribution between 17 groups with *Bacteroidota*-enriched cecal microbiomes and 19 groups with *Bacillota* + ACT cecal microbiomes (Figure 6).

Comparison of *Bacteroidota*-enriched and *Bacillota* + ACT cecal microbiomes in the top 20 genera distribution revealed the predominance of *Bacteroides* and *Alistipes* (belonging to the phylum *Bacteroidota*), as well as *Lactobacillus* (belonging to phylum *Bacillota*) in the F1(−) microbiomes, while in F1(+) microbiomes there was a high content of *Butyricicoccus* and 12 other taxa belonging exclusively to the phylum *Bacillota* (Table 3). Against this background, four numerous genera (*Faecalibacterium*, *Ruminococcus*, *Fusicatenibacter* and unclassified members of the family *Lachnospiraceae*), also belonging to the phylum *Bacillota*, did not show statistically significant differences in abundance between F1(−) and F1(+) microbiomes.

### 3.5. Microbiome–Elementome Associations

Associations between the cecal microbiome compositions and the elemental compositions of broiler chickens biosamples were assessed in two aspects: (i) all liver and meat samples belonging to *Bacteroidota*-enriched F1(−) and *Bacillota* + ACT F1(+) cecal microbiome patterns were compared for their content of 25 elements and (ii) complete data sets were distributed according to microbiome and element definitions.

As shown in Table 4, a contrasting distribution of most of the analyzed elements was established between groups belonging to the F1(−) and F1(+) microbiome patterns. All biosamples of chickens belonging to the *Bacteroidota*-enriched cecal microbiome were characterized by a significantly (*p* < 0.001) increased content of all determined macro elements (Na, K, Mg, Ca, P) and a number of trace elements (I, Mn and Cu). In addition, increased concentrations of Sr, Se, Fe, Co and Zn, as well as some toxic elements (Al, Pb and Ni), were found in liver samples related to the F1(−) pattern, while thigh samples were enriched in Sn and As and breast samples in As and Cd. In contrast, chicken biosamples belonging to the *Bacillota* + ACT cecal microbiome showed decreased levels of a number of the macro, micro and toxic elements listed above but were enriched in Si, while liver samples also showed a high V content and meat showed a high Cr content.

In the next step, the distribution of complete data sets was analyzed via their projection onto the F1 microbiome × F1 elementome factor plane. As shown in Figure 7, in chicken liver samples, a clear relationship was established between the previously described *Bacteroidota*-enriched and *Bacillota* + ACT cecal microbiome and macro, trace and toxic element compositions, designated as α-, β- and γ-elementomes. All α-elementomes (enriched with macro elements Na, K, Mg, Ca, P; majority trace elements—Sr, Se, Mn, Fe, Co, Cu, Zn; and some toxic elements—B, Pb, Ni, Cd) and γ-elementomes (increased in Li, Al, As and Hg content) were strictly related to *Bacteroidota*-enriched microbiomes. In turn, most β-elementomes demonstrated decreases in the contents of most macro, trace and toxic elements but were enriched in Si, V and Cr, associated with *Bacillota* + ACT microbiomes, while several groups characterized by F1_MB_ values from 0 to −1 were formally related to the F1(−) microbiome pattern. Very similar distributions were also stated for meat samples where minor variations were determined by differences in the thigh and breast elementomes, optionally including or excluding some trace and toxic elements.

### 3.6. Bacterial Taxa Influenced the Elemental Compositions in Broiler Chicken Biosamples

To identify bacterial taxa that most significantly influence the accumulation of elements in broiler chicken biosamples, the effects of the phyla included in the microbiome pattern structure, as well as the genera that shape the caecum microbiome biodiversity, were analyzed.

Table 5 presents the coefficient of determination R^2^ values, characterizing the significant (*p* < 0.001) influence of bacterial phyla abundance on the macro, trace and toxic element accumulation in liver (L), thigh (T) and breast (B) samples. Numerous effects on the content of most macro elements (Na, K, Mg, Ca, P), a number of trace elements (I, V, Cr, Mn) and single toxic elements (Pb, As) were found for the phyla *Bacteroidota* and *Bacillota*, which were alternative (positive or negative) in this influence. However, these data only partially supported the holistic effects of *Bacteroidota*-enriched and *Bacillota* + ACT microbiomes, involving fewer elements and biosamples. In turn, the *Cyanobacteriota* phylum, part of the *Bacillota* + ACT pattern, was associated with a negative influence on K (in the thigh) and As (in the thigh and breast) and a positive influence on the Si content in liver samples. Two other phyla (*Actinomycetota* and *Thermodesulfobacteriota*), also included in the *Bacillota* + ACT pattern, were associated with Fe accumulation in the breast, which was complemented by the effect of *Actinomycetota* on the V content in the liver and *Thermodesulfobacteriota* on the Ni content in the thigh samples. Moreover, we report a dispersion of the associations to other bacterial phyla, among which only *Verrucomicrobiota* did not influence the accumulation of any detectable elements.

The influence of individual bacterial genera presented in the caecum microbiomes on the accumulation of elements in broiler chicken biosamples was visualized in a 2-dimensional heat map, representing the magnitude of the association of individual values within a data set as a color. According to this approach, the raw data were clustered and three robust clusters were found for both bacterial taxa abundance and element content.

In the liver samples (Figure 8), the clusters of elements were as follows: (i) a composition of macro elements (Na, K, Mg, Ca, P) with majority trace elements (Sr, Se, Mn, Fe, Co, Cu, Zn) and some toxic elements (B, Sn, Pb, Ni, Cd), which corresponded well with the α-elementome; (ii) a cluster of Li, I and three toxic elements, As, Al, Hg, similar to the γ-elementome; (iii) Si, V and Cr compositions equivalent to the β-elementome. In turn, the bacterial genera showed clusters corresponding to the preferential accumulation or exclusion of elements present in the clusters (elementomes) described above. The bacterial cluster associated with the accumulation of most macro, trace and toxic elements included 47 taxa (40 classified at the genus level and 7 at the family level), among which the most abundant genus was *Bacteroides*, while the top 10 significant influencers were *Acetivibrio*, *Anaerobacterium*, *Anaerovorax*, *Clostridium IV*, *Coprobacter*, *Dorea* and *Lactobacillus* (belonging to the phylum *Bacillota*) and *Bifidobacterium* (phylum *Actinomycetota)*, *Ralstonia* (phylum *Pseudomonadota*) and *unclassified Rikenellaceae* (phylum *Bacteroidota*), with each having an r value more than 0.7 (*p* < 0.001) for at least 10 elements. The cluster associated with the accumulation of Li, I. As, Al and Hg consisted of 17 genera and 4 bacterial taxa classified to the family level, including *Alistipes* (the most abundant genus in the cluster) and *Odoribacter*, both belonging to the phylum *Bacteroidota*, as well as *Anaerofilum*, *Blautia*, *Erysipelotrichaceae Incertae sedis*, *Hespellia*, *Lachnospiracea incertae sedis* (phylum *Bacillota*), *Parasutterella* (phylum *Pseudomonadota*), unclassified members of the families *Coriobacteriaceae* (phylum *Actinomycetota*) and *Desulfovibrionaceae* (phylum *Thermodesulfobacteriota*), which significantly increased the content of at least three of the five elements. The most numerous bacterial cluster was associated with the accumulation of Si, V and Cr and included 71 taxa; among them, *Butyrococcus* was the most abundant, and *Dielma*, *Dysosmobacter*, *Frisingicoccus* and unclassified *Erysipelotrichaceae* all belonging exclusively to the phylum *Bacillota* were the most significant influencers, while most of the other genera changed the content of only one named element. Another closely related subcluster was indifferent to the accumulation of elements and consisted of eight bacterial taxa, including the genera *Anaerostipes*, *Caecibacterium*, *Clostridium sensu stricto*, *Clostridium XlVa*, *Coprococcus*, *Eggerthella*, *Faecalibacterium* and unclassified members of the family *Enterobacteriaceae*.

A similar three-cluster structure was found in heat maps based on broiler chicken meat data sets, with toxic element clusters missing Al and I (thigh) or As, Al and I (breast) moving to macro, trace and toxic element clusters. Against this background, most of the influencing genera confirmed their bioactivity, although they were associated with the accumulation of a few elements.

## 4. Discussion

This study presents the results of the determination of 25 elements in broiler chicken liver and meat samples, which are in good agreement with previously published data worldwide [6,7,40] and show these products as the sources of macro and essential trace elements for balancing body function in humans. We also report a preferential accumulation of Na, Ca and P in liver samples vs. meat samples, which was accompanied by an increased content of a number of essential trace elements (Se, V, Co, Cu, Zn, and especially Mn and Fe), while the thigh and breast were characterized by increased K and Mg content compared to the liver. In turn, compliance according to FAO/WHO standards [4] showed a moderate excess of Mn, Cu and Zn concentrations (in addition, Cr in the liver and Se in the meat), while most toxic elements (B, Al, Sn, As, Cd, Hg) were below the maximum permissible levels, and the content of single heavy metals (Pb and Ni) was only sporadically exceeded, which indicates the safety of the analyzed biosamples for human health.

Notably, variations in elemental concentrations in broiler chicken liver and meat samples were highly correlated and stratified, with three typical compositions termed “elementomes” and defined as the content of all (or at least most) macro, trace and toxic elements [39]. The observed α-elementome was characterized by a high content of macro elements (Na, K, Mg, Ca, P), majority trace elements (Sr, Se, Mn, Fe, Co, Cu, Zn) and some toxic elements (B, Pb, Ni, Cd). The alternative β-elementome showed high accumulation of Si, V and Cr with decreased concentrations of most of the macro, trace and toxic elements mentioned above. In turn, the γ-elementome has an increased content of Li and a number of toxic elements (Al, As and Hg). In controlled experiments, these variations could not be explained by differences in host genetics or dietary supplementation; therefore, our attention was focused on the gut microbiome, which plays a significant role in macro, trace and toxic element metabolism in animals [19] and humans [22].

An in-depth characterization of broiler chickens’ cecal microbiome revealed nine bacterial phyla, divided into 16 classes, 26 orders, represented by 62 families and, at the lowest taxonomic level, including 130 genera. Following analyses of the microbiome structure, two distinct enterotypes, designated “microbiome patterns”, were determined [33]: the first is enriched in the phylum *Bacteroidota* and the second is dominated by *Bacillota* and coupled with members of the phyla *Actinomycetota*, *Cyanobacteriota* and *Thermodesulfobacteriota.* Similar microbiome compositions (cecal enterotypes), showing significant differences in the *Firmicutes*/*Bacteroidetes* (currently *Bacillota*/*Bacteroidota*) ratio, were described by Hay et al. [41] and Marcolla et al. [42], which supports the alternative abundance of bacterial phyla listed above. At a low taxonomic level, these microbiome patterns also differed in the core genera distribution, with *Bacteroides* and *Alistipes* predominant in *Bacteroidota*-enriched communities and *Butyricicoccus* and other clostridial taxa predominant in *Bacillota* + ACT communities.

Comparing the cecal microbial community and the elemental composition of the broiler chicken biosamples, clear correspondences were shown. Multiple significant differences in the content of most of the determined elements were found between the groups belonging to the *Bacteroidota*-enriched and *Bacillota* + ACT microbiomes. The first microbiome pattern was characterized by an increased content of all determined macro elements (Na, K, Mg, Ca, P), a number of trace elements (I, Mn, Cu, etc.) and some toxic elements. The second pattern showed a decreased content of numerous elements listed above but was enriched in Si and optionally in V and Cr. A direct comparison of microbiomes and elementomes demonstrated a clear correspondence between the α- and γ-elementomes belonging to the *Bacteroidota*-enriched pattern, while the β-elementome was predominantly found in groups belonging to the *Bacillota* + ACT pattern. In our opinion, the established features are a novel example of genuine discreteness in biology, where data sets vary discretely, not continuously [43].

To identify the bacterial taxa that most significantly influence the elemental content in broiler chicken biosamples, an analysis of microbiomes and elementome associations in the types and genera definitions was carried out. At the phylum level, alternative effects were confirmed for *Bacteroidota* and *Bacillota*, which are the core phyla in *Bacteroidota*-enriched and *Bacillota* + ACT- microbiomes, but they did not completely explain the element distribution. In turn, at the genus level, we first describe three bacterial clusters strictly associated with the element accumulation represented by the α-, β- and γ-elementomes. The bacterial cluster associated with the α-elementome included 47 taxa, where the most abundant genus was *Bacteroides*; the bacterial cluster associated with the β-elementome included 71 taxa, among which the genus *Butyrococcus* belonging to the phylum *Bacillota* was the most abundant; and the γ-elementome was related to the bacterial cluster dominated by the genus *Alistipes* belonging to the phylum *Bacteroidota* and included 21 taxa. Notably, each bacterial cluster included genera belonging to different phyla found in chicken cecal microbiomes, suggesting similar effects on elemental metabolism in phylogenetically distinct bacterial taxa.

Some of the bacterial genera identified in the above clusters have already been described to influence elemental metabolism. The most information has been collected about *Lactobacillus*, belonging to the phylum *Bacillota*, which included the top 20 most abundant genera and is related to the α-elementome, which modulates Mn, Fe and Zn accumulation via metal requirements and metallotransporter function [44,45]. In addition, *Lactobacillus* controls Fe metabolism by excreting p-hydroxyphenyllactic acid, which causes the reduction of Fe(III) to Fe(II) and is essential for iron absorption in the gut [46]. *Lactobacillus* and *Bifidobacterium* (belonging to the phylum *Actinomycetota*, related to the α-elementome) also increased Ca accumulation [47] by altering calcium solubility and short-chain fatty acids production, which might improve protein digestibility and lead to increased calcium release and absorption [48]. *Bifidobacterium*-caused gut acidification is also beneficial for Mg absorption [49]. The current understanding of the gut microbiota’s role in Fe, Mn, Zn and Cu metabolism is summarized in the reviews by Pajarillo et al. [19] and Ma et al. [50]; however, the available data relate little to the bacterial taxa that are dominant in cecal microbiomes and do not completely explain the results of this study.

In discussing the current results, some other methodological limitations should be noted, including the quantity and diversity of data analyzed. First, the study was performed using only two chicken crosses, limiting genetic variability, which is important for chicken metabolism [9] and has not yet been characterized in terms of the impact on gut microbiome diversity. Secondly, the cecal microbiome was assessed at the “end point” (post-slaughter) only, while the temporal dynamics of the chicken gut microbial community are well described [51]. For this reason, it is unclear when the described microbiome patterns emerged and when they began to influence the elements’ accumulation in the chicken body, which requires further studies with improved experimental design. Thirdly, the data collection did not take into account the chickens’ health, although the role of macro, trace and toxic elements in birds’ physiology is well established [52,53]. Accordingly, our next studies will target this goal, focusing on the inflammation status and growth performance of chickens with different gut microbiome patterns and body elemental compositions.

In sum, the obtained results provide empirical data on the relationship between the chickens’ gut microbial community and the elemental composition in the host body, which opens up prospects for an in-depth analysis of this phenomenon [54]. In the future, it will be interesting to see how and which diet additives influence the gut microbiome, which in turn impacts the absorption and accumulation of macro, trace and toxic elements [55]. It is also important to analyze the effect of dietary supplements on gut development and function, which is also involved in the absorption and accumulation of elements [56,57]. This integrated approach appears to be a very promising method for the selection of optimal dietary supplements for poultry farming, taking into account their impact on the gut microbiome, mineral, protein and fat metabolism, inflammatory status and growth performance.

## 5. Conclusions

The obtained results indicate multiple strong associations between the gut microbial community and the elemental body composition of broiler chickens, where the microbiomes’ discreteness, represented in the “microbiome patterns” concept, finds a correspondence in the discrete elemental compositions of chicken tissues, designated as “elementomes”. In turn, the finding of certain bacterial clusters and individual bacterial taxa positively or negatively associated with the macro, trace and toxic element content in broiler chickens suggests that specific gut microbes and microbial communities might modulate the accumulation or detoxification of elements. This insight proposes a novel strategy to improve the deficiency or excess of certain elements in the host by modulation of gut bacterial compositions, which needs to be verified with further in vivo experiments.

## Figures and Tables

**Figure 1 biology-13-00975-f001:**
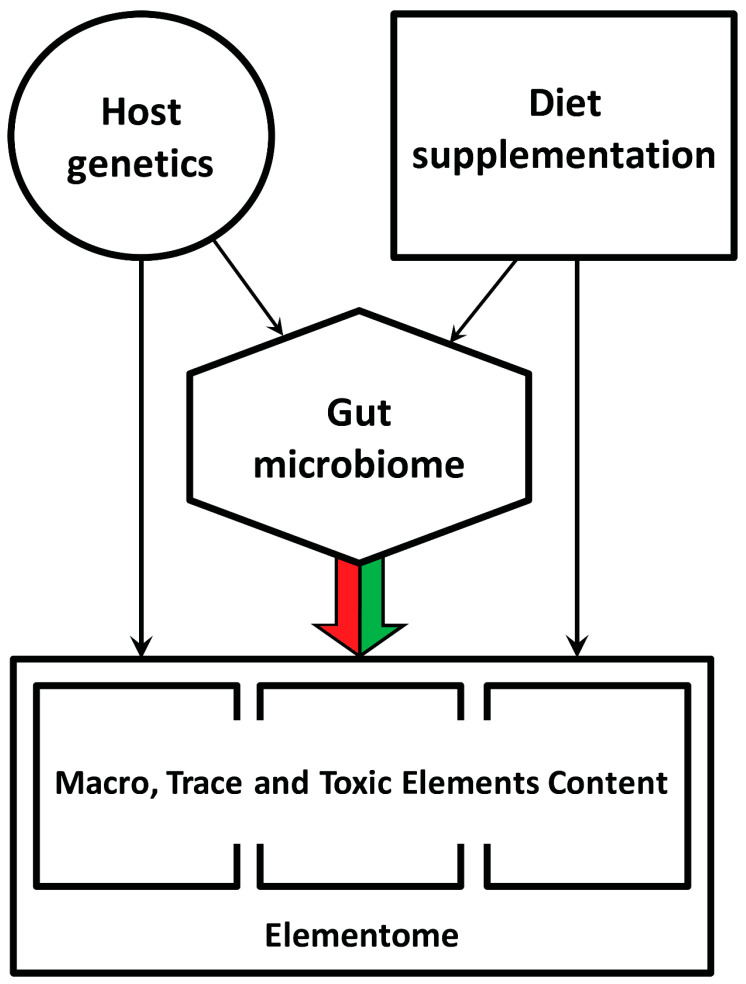
General overview of factors influencing the macro, trace and toxic element content in the chicken body. The colored arrow indicates the present study goal.

**Figure 2 biology-13-00975-f002:**
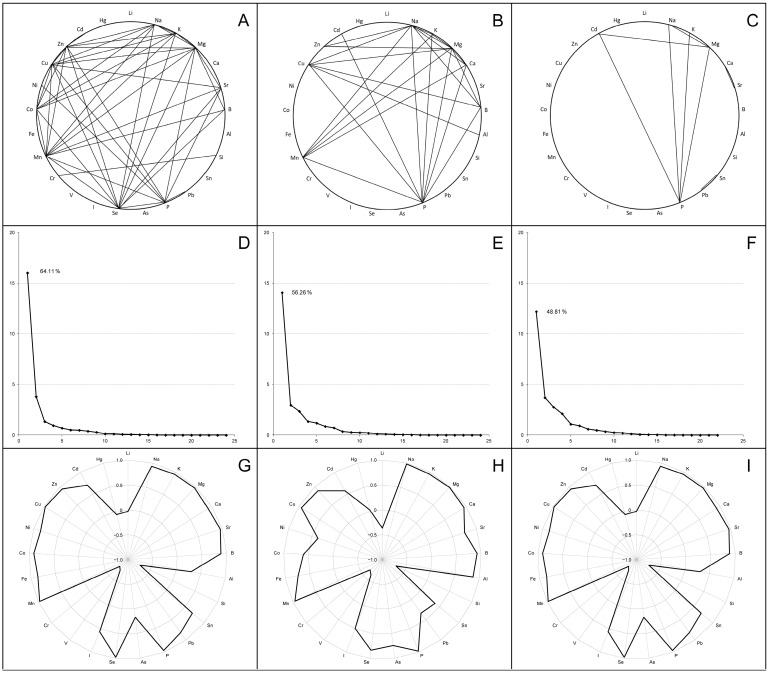
Correlation analysis (**A**–**C**), principal component analysis (**D**–**F**) and first component structure (**G**–**I**) among the examined elements in the livers (**A**,**D**,**G**), thigh muscles (**B**,**E**,**H**) and breast muscles (**C**,**F**,**I**) in broiler chickens. Parts (**A**–**C**) indicate the strong positive correlations between elements (r ≥ 0.9). Parts (**D**–**F**) present scree plots where the *x*-axis is the component number and the *y*-axis is the component eigenvalue. Parts (**G**–**I**) show factor loadings (between +1.0 and −1.0) for examined elements in the first principal component (F1).

**Figure 3 biology-13-00975-f003:**
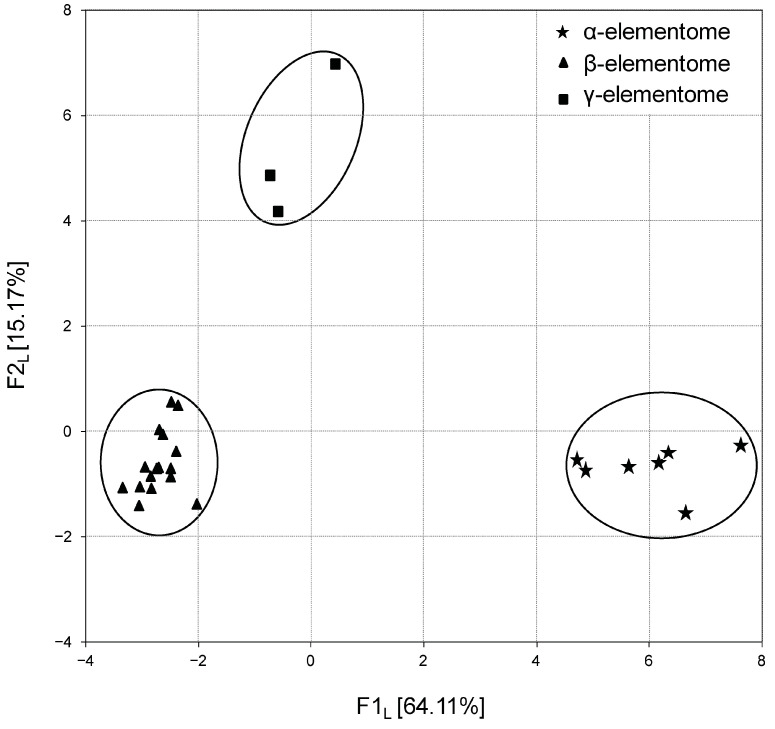
Projection of the macro, trace and toxic element compositions (elementomes) in the liver of broiler chickens on the F1_L_ × F2_L_ factor plane, estimated using principal component analysis. α-elementomes marked with asterisks accumulate Na, K, Mg, Ca, P, Sr, Se, Mn, Fe, Co, Cu, Zn, B, Pb, Ni and Cd; β-elementomes marked by triangles show high contents of Si, V and Cr; γ-elementomes marked by squares are characterized by increased concentrations of Li, Al, As and Hg.

**Figure 4 biology-13-00975-f004:**
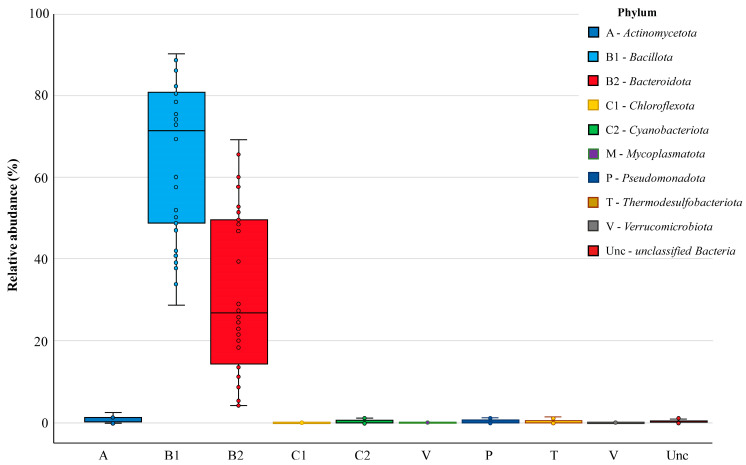
The relative abundance of bacterial phyla in chicken cecal microbiomes. Box plots show the distribution of phyla abundance by displaying the data quartiles.

**Figure 5 biology-13-00975-f005:**
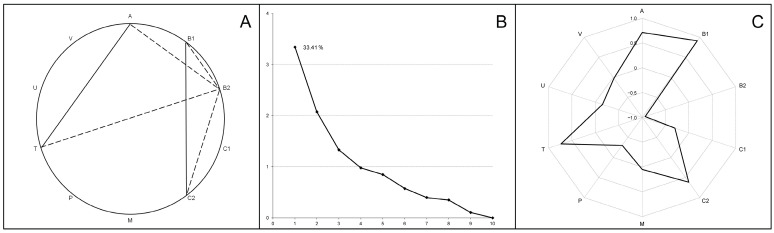
Correlation analysis (**A**), principal component analysis (**B**) and first component structure (**C**) characterizing the cecal microbiome composition in broiler chickens. Part A shows statistically significant (*p* < 0.001) bacterial phyla co-occurrence, with positive correlations in solid lines and negative correlations in dotted lines. Part B presents scree plots where the *x*-axis is the component number and the *y*-axis is the component eigenvalue. Part C indicates factor loadings (between +1.0 and −1.0) for occurred bacterial phyla in the first principal component. Bacterial phyla designations: A—*Actinomycetota*; B1—*Bacillota*; B2—*Bacteroidota*; C1—*Chloroflexota*; C2—*Cyanobacteriota*; M—*Mycoplasmatota*; P—*Pseudomonadota*; T—*Thermodesulfobacteriota*; V—*Verrucomicrobiota*; Unc—*unclassified Bacteria*.

**Figure 6 biology-13-00975-f006:**
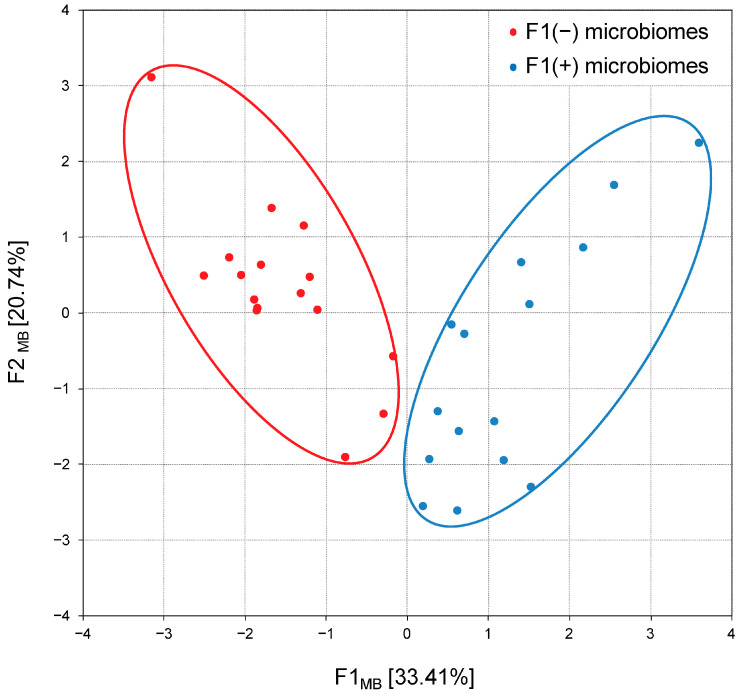
Projection of the cecal microbiome compositions (patterns) in broiler chickens on the F1_MB_ × F2_MB_ factor plane, assessed using principal component analysis. F1(−) microbiomes represent *Bacteroidota*-enriched cecal microbial communities; F1(+) microbiomes represent *Bacillota* + ACT (*Actinomycetota*, *Cyanobacteriota* and *Thermodesulfobacteriota*) cecal microbial communities.

**Figure 7 biology-13-00975-f007:**
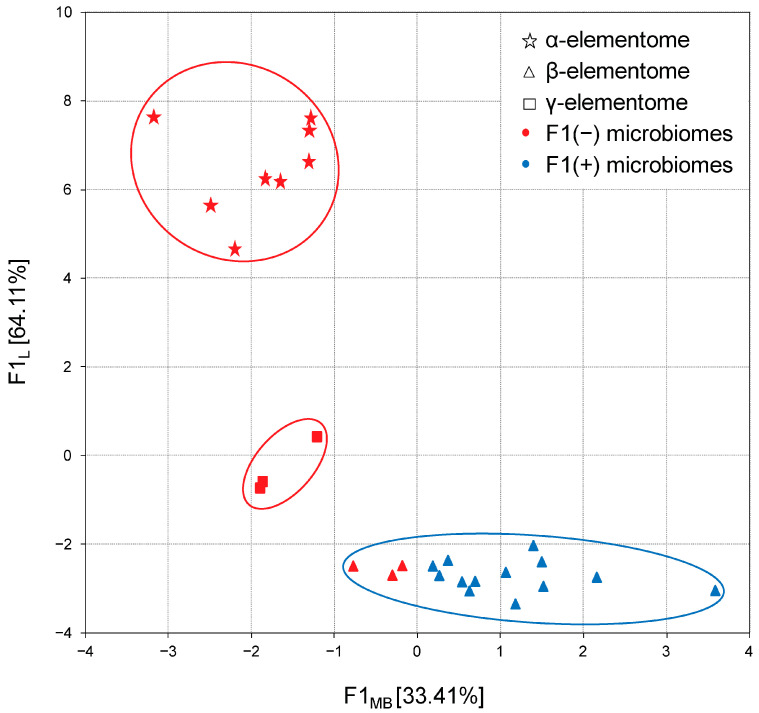
Projection of the macro and trace element compositions (elementomes) in the liver of broiler chickens and the cecal compositions (microbiomes) on the F1_L_ × F1_MB_ factor plane. α-elementomes marked with asterisks accumulate Na, K, Mg, Ca, P, Sr, Se, Mn, Fe, Co, Cu, Zn, B, Pb, Ni and Cd; β-elementomes marked by triangles show high Si, V and Cr content; γ-elementomes marked by squares are characterized by increased concentrations of Li, Al, As and Hg. F1(−) microbiomes represent *Bacteroidota*-enriched cecal microbial communities; F1(+) microbiomes represent *Bacillota* + ACT (*Actinomycetota*, *Cyanobacteriota* and *Thermodesulfobacteriota*) cecal microbial communities.

**Figure 8 biology-13-00975-f008:**
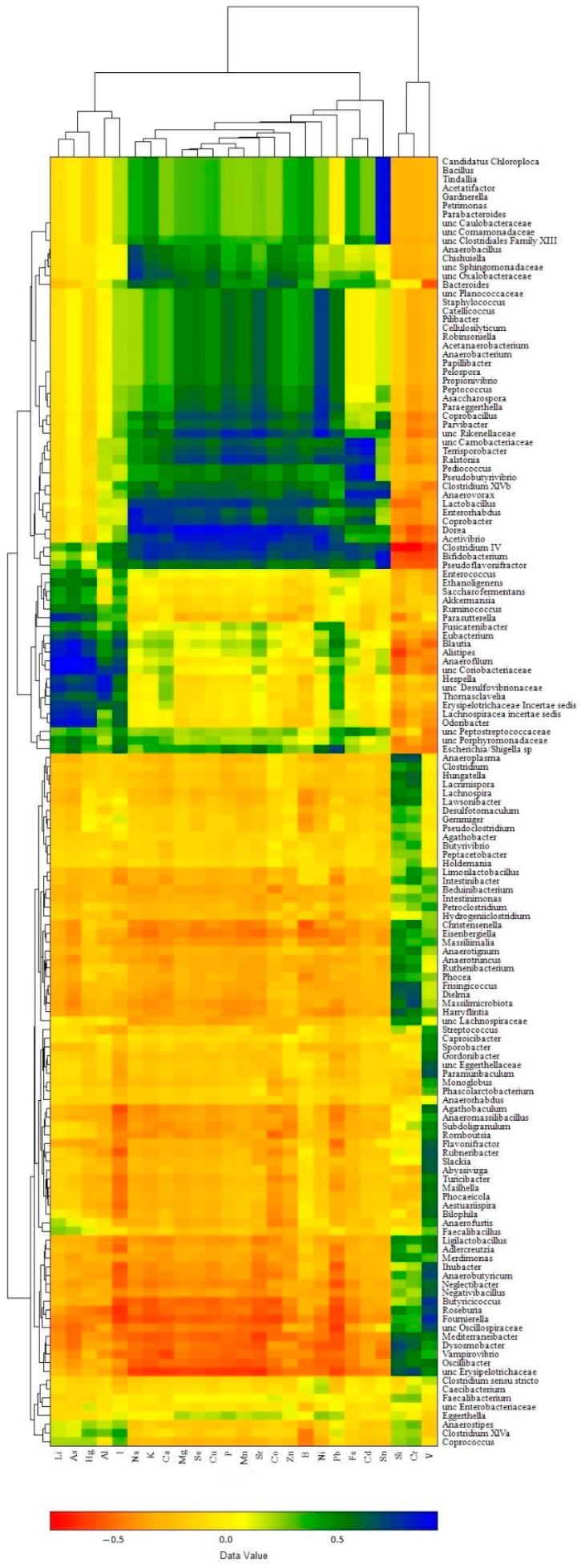
Clustered heat map showing associations between bacterial genera abundance in chicken cecal microbiomes and element accumulation in liver samples.

**Table 1 biology-13-00975-t001:** The content of macro, trace and toxic elements in broiler chicken biosamples compared with the FAO/WHO maximum permitted concentration of the element in poultry products.

Element	Liver	Thigh	Breast	Maximum Permissible Level (FAO/WHO)	
				Liver	Meat
Macro elements (‰)					
Na	1.11 [0.95–3.16]	0.79 [0.69–2.82] *	0.58 [0.49–1.33] *	NA	NA
K	3.04 [2.72–7.17]	4.27 [4.01–9.34] *	4.40 [4.00–7.37] *	NA	NA
Mg	0.24 [0.21–0.58]	0.31 [0.29–1.08] *	0.34 [0.32–0.97] *	NA	NA
Ca	0.08 [0.07–0.22]	0.06 [0.05–0.18] *	0.07 [0.05–0.16]	NA	NA
P	3.42 [3.09–7.70]	2.19 [2.09–6.27] *	2.48 [2.31–7.72] *	NA	NA
Essential trace elements (ppm)					
Li	0.01 [0.00–0.01]	0.01 [0.00–0.01]	0.01 [0.00–0.01]	NA	NA
Sr	0.09 [0.06–0.27]	0.05 [0.04–0.16]	0.07 [0.05–0.28]	NA	NA
Si	16.72 [6.24–33.97]	24.72 [6.40–29.70]	22.16 [10.40–26.99]	NA	NA
Se	0.77 [0.67–1.98]	0.19 [0.18–0.48] *	0.18 [0.15–0.40] *	NA	0.5
I	0.12 [0.05–0.20]	0.08 [0.02–0.13]	0.12 [0.03–0.29]	NA	NA
V	0.04 [0.01–0.07]	0.01 [0.01–0.02] *	0.01 [0.01–0.02] *	NA	NA
Cr	0.27 [0.17–0.39]	0.26 [0.12–0.31]	0.29 [0.15–0.32]	0.05	1.00
Mn	3.40 [2.98–8.59]	0.21 [0.18–0.57] *	0.21 [0.17–0.57] *	0.50	0.50
Fe	291.83 [232.20–656.22]	19.72 [13.02–36.76] *	25.84 [14.28–34.98] *	NA	NA
Co	0.02 [0.02–0.04]	0.00 [0.00–0.01] *	0.00 [0.00–0.01] *	NA	NA
Cu	4.49 [3.92–11.25]	0.66 [0.49–2.11] *	0.46 [0.42–1.08] *	1.00	1.00
Zn	36.41 [29.68–92.26]	19.24 [16.95–53.47] *	19.53 [9.02–24.57] *	20.00	20.00
Toxic elements (ppm)					
B	0.21 [0.19–0.46]	0.16 [0.11–0.27]	0.17 [0.13–0.33]	NA	10.00
Al	1.09 [0.86–1.37]	1.35 [0.90–3.17]	1.91 [1.28–3.61] *	NA	1.00
Sn	0.01 [0.01–0.02]	0.00 [0.00–0.01]	0.01 [0.01–0.02]	NA	NA
Pb	0.02 [0.01–0.03]	0.02 [0.01–0.02]	0.02 [0.01–0.03]	0.10	0.10
As	0.01 [0.00–0.01]	0.00 [0.00–0.01]	0.00 [0.00–0.01]	NA	0.10
Ni	0.04 [0.02–0.21]	0.03 [0.01–0.06]	0.04 [0.02–0.06]	NA	0.50
Cd	0.03 [0.02–0.19]	0.00 [0.00–0.00] *	0.00 [0.00–0.04] *	0.50	0.05
Hg	0.00 [0.00–0.01]	0.00 [0.00–0.01]	0.01 [0.00–0.01]	NA	NA

Note: data are presented as median value [Q1–Q3]. * *p* < 0.001 between meat and liver by Mann–Whitney U test. NA—not available.

**Table 2 biology-13-00975-t002:** Top 20 bacterial genera in chicken cecal microbiomes.

Genera	Occurrence (%)	Abundance (%) *
Belonging to the phylum *Bacteroidota*		
*Bacteroides*	100	8.04 [0.22–24.75]
*Alistipes*	72.22	6.77 [0.0–27.69]
Belonging to the phylum *Bacillota*		
*Butyricicoccus*	100	0.93 [0.34–1.39]
*Faecalibacterium*	100	4.93 [3.02–7.16]
*Lactobacillus*	100	2.79 [0.93–4.32]
*Ruminococcus*	100	0.58 [0.28–1.47]
*uncassified Lachnospiraceae*	100	3.59 [3.14–4.35]
*unclassified Oscillospiraceae*	100	17.40 [11.87–21.44]
*Christensenella*	97.22	1.44 [0.12–3.49]
*Fusicatenibacter*	97.22	0.40 [0.12–0.55]
*Romboutsia*	97.22	0.45 [0.15–0.90]
*Subdoligranulum*	97.22	1.02 [0.38–1.78]
*Eisenbergiella*	94.44	1.17 [0.15–2.98]
*Turicibacter*	88.89	0.35 [0.04–2.35]
*Flavonifractor*	77.78	0.42 [0.03–0.64]
*Intestinimonas*	72.22	0.39 [0.0–1.69]
*Anaerotignum*	63.89	0.59 [0.0–1.04]
*Dysosmobacter*	61.11	0.54 [0.0–1.10]
*Mediterraneibacter*	61.11	0.99 [0.0–2.16]
*Neglectibacter*	61.11	0.34 [0.0–0.59]

Note: * Presented as median value [Q1–Q3].

**Table 3 biology-13-00975-t003:** Distribution of top 20 bacterial genera between the *Bacteroidota*-enriched F1(−) and *Bacillota* + ACT F1(+) cecal microbiomes.

Genera	Abundance in F1(−) Microbiomes, %	Abundance in F1(+) Microbiomes, %	*p* Value
*Bacteroides*	13.72 [4.61–35.68]	0.48 [0.20–22.87]	2.7 × 10^−2^
*Alistipes*	35.95 [12.66–44.73]	2.94 [0–8.08]	1.2 × 10^−4^
*Butyricicoccus*	0.26 [0.18–0.53]	1.28 [0.93–1.54]	4.6 × 10^−5^
*Faecalibacterium*	5.02 [4.65–7.96]	4.26 [2.64–5.45]	>0.05
*Lactobacillus*	4.24 [3.04–6.72]	1.57 [0.57–2.99]	3.1 × 10^−3^
*Ruminococcus*	0.93 [0.35–1.53]	0.38 [0.19–0.99]	>0.05
*uncassified Lachnospiraceae*	3.31 [1.68–3.89]	3.80 [3.22–4.53]	>0.05
*unclassified Oscillospiraceae*	11.54 [7.97–14.67]	20.10 [17.45–21.44]	2.9 × 10^−4^
*Christensenella*	0.06 [0.02–0.20]	3.09 [1.44–4.62]	9.4 × 10^−6^
*Fusicatenibacter*	0.60 [0.05–0.87]	0.32 [0.14–0.50]	>0.05
*Romboutsia*	0.07 [0.01–0.32]	0.62 [0.44–1.39]	2.2 × 10^−4^
*Subdoligranulum*	0.37 [0.21–0.65]	1.71 [0.94–2.38]	1.5 × 10^−3^
*Eisenbergiella*	0.09 [0.01–0.43]	2.69 [1.69–4.94]	4.4 × 10^−6^
*Turicibacter*	0.03 [0–0.08]	1.22 [0.41–6.20]	2.7 × 10^−6^
*Flavonifractor*	0.09 [0.01–0.42]	0.51 [0.36–0.70]	4.9 × 10^−2^
*Intestinimonas*	0.01 [0.0–0.04]	1.24 [0.88–2.45]	2.7 × 10^−3^
*Anaerotignum*	0.0 [0.0–0.0]	0.68 [0.59–1.44]	1.4 × 10^−4^
*Dysosmobacter*	0.0 [0.0–0.0]	0.99 [0.89–1.68]	4.1 × 10^−7^
*Mediterraneibacter*	0.0 [0.0–0.0]	2.12 [1.19–2.92]	9.8 × 10^−7^
*Neglectibacter*	0.0 [0.0–0.0]	0.49 [0.37–0.71]	1.6 × 10^−6^

Note: data are presented as median value [Q1–Q3]. *p* values between groups were calculated using the Mann–Whitney U test.

**Table 4 biology-13-00975-t004:** Distribution of macro, trace and toxic elements in the liver, thigh and breast of broiler chickens belonging to the *Bacteroidota*-enriched F1(−) and *Bacillota* + ACT F1(+) cecal microbiome patterns.

Element	Liver			Thigh			Breast		
	F1(−) Microbiome Group	F1(+) Microbiome Group	*p* Value	F1(−) Microbiome Group	F1(+) Microbiome Group	*p* Value	F1(−) Microbiome Group	F1(+) Microbiome Group	*p* Value
Macro elements (‰)									
Na	3.36 [1.56–3.81]	1.02 [0.87–1.02]	3.3 × 10^−5^	2.85 [0.84–4.04]	0.71 [0.66–0.71]	1.6 × 10^−4^	1.38 [0.83–1.58]	0.50 [0.48–0.50]	4.0 × 10^−6^
K	7.17 [4.01–8.53]	2.73 [2.67–2.73]	1.6 × 10^−5^	9.42 [6.11–11.05]	4.10 [3.81–4.10]	1.0 × 10^−4^	7.37 [6.22–7.58]	4.10 [3.79–4.10]	1.6 × 10^−4^
Mg	0.59 [0.25–0.75]	0.22 [0.21–0.22]	3.8 × 10^−5^	1.10 [0.36–1.17]	0.30 [0.29–0.30]	6.0 × 10^−6^	0.98 [0.40–1.05]	0.33 [0.30–0.33]	8.4 × 10^−5^
Ca	0.22 [0.11–0.28]	0.08 [0.07–0.08]	5.0 × 10^−6^	0.19 [0.07–0.22]	0.05 [0.04–0.05]	1.0 × 10^−4^	0.16 [0.07–0.21]	0.06 [0.05–0.06]	6.6 × 10^−4^
P	8.03 [3.94–11.66]	3.24 [3.00–3.24]	1,1 × 10^−4^	6.38 [2.91–7.20]	2.10 [2.04–2.10]	1.1 × 10^−5^	7.72 [3.17–8.24]	2.35 [2.01–2.35]	4.3 × 10^−5^
Essential trace elements (ppm)									
Li	0.01 [0.00–0.05]	0.01 [0.00–0.01]	>0.05	0.00 [0.00–0.02]	0.01 [0.01–0.01]	>0.05	0.00 [0.00–0.01]	0.01 [0.00–0.01]	>0.05
Sr	0.28 [0.12–0.38]	0.06 [0.05–0.06]	1.2 × 10^−4^	0.20 [0.05–0.32]	0.05 [0.05–0.05]	>0.01	0.26 [0.06–0.29]	0.06 [0.05–0.06]	>0.05
Si	6.15 [4.42–7.03]	29.32 [17.53–29.32]	1.4 × 10^−4^	5.70 [4.62–10.04]	28.53 [24.79–28.53]	8.5 × 10^−5^	10.40 [7.59–13.70]	24.60 [22.63–24.60]	2.1 × 10^−4^
Se	2.24 [0.80–2.47]	0.70 [0.67–0.70]	3.0 × 10^−5^	0.51 [0.22–0.52]	0.19 [0.17–0.19]	>0.001	0.40 [0.24–0.45]	0.16 [0.14–0.16]	>0.001
I	0.20 [0.18–0.29]	0.05 [0.04–0.05]	4.0 × 10^−6^	0.13 [0.12–0.16]	0.02 [0.02–0.02]	8.0 × 10^−5^	0.30 [0.16–0.66]	0.04 [0.01–0.04]	1.6 × 10^−5^
V	0.01 [0.01–0.04]	0.05 [0.05–0.05]	1.7 × 10^−4^	0.01 [0.01–0.01]	0.02 [0.02–0.02]	>0.001	0.01 [0.01–0.02]	0.01 [0.01–0.01]	>0.05
Cr	0.17 [0.09–0.37]	0.32 [0.27–0.32]	>0.01	0.12 [0.10–0.16]	0.31 [0.29–0.31]	6.0 × 10^−6^	0.14 [0.11–0.27]	0.32 [0.30–0.32]	4.3 × 10^−4^
Mn	10.17 [3.90–13.18]	3.13 [2.70–3.13]	1.6 × 10^−4^	0.59 [0.33–0.68]	0.20 [0.18–0.20]	4.7 × 10^−4^	0.57 [0.28–0.61]	0.19 [0.16–0.19]	6.8 × 10^−4^
Fe	833.00 [416.00–1342.00]	234.50 [213.25–234.50]	1.2 × 10^−5^	33.94 [14.25–42.83]	17.27 [12.84–17.27]	>0.01	24.20 [17.20–27.98]	26.34 [13.80–26.34]	>0.05
Co	0.05 [0.02–0.05]	0.02 [0.01–0.02]	2.5 × 10^−4^	0.01 [0.00–0.01]	0.00 [0.00–0.00]	>0.001	0.01 [0.01–0.01]	0.00 [0.00–0.00]	>0.001
Cu	12.01 [4.72–13.55]	3.98 [3.83–3.98]	2.9 × 10^−5^	2.16 [0.72–2.57]	0.57 [0.49–0.57]	5.4 × 10^−4^	1.08 [0.49–1.39]	0.43 [0.36–0.43]	1.5 × 10^−4^
Zn	94.24 [41.40–114.00]	30.88 [25.68–30.88]	3.9 × 10^−4^	54.30 [14.90–57.87]	18.31 [17.36–18.31]	>0.01	24.49 [9.44–26.99]	14.20 [8.84–14.20]	>0.01
Toxic elements (ppm)									
B	0.48 [0.20–0.49]	0.20 [0.19–0.20]	>0.01	0.50 [0.17–0.66]	0.15 [0.12–0.15]	>0.001	0.41 [0.14–0.55]	0.17 [0.13–0.17]	>0.01
Al	1.32 [1.14–1.66]	0.86 [0.77–0.86]	2.2 × 10^−4^	3.55 [1.14–5.41]	1.21 [0.91–1.21]	>0.01	3.71 [1.24–5.13]	1.71 [1.32–1.71]	>0.05
Sn	0.02 [0.01–0.04]	0.01 [0.01–0.01]	>0.001	0.01 [0.01–0.02]	0.00 [0.00–0.00]	8.1 × 10^−4^	0.02 [0.01–0.03]	0.01 [0.01–0.01]	>0.01
Pb	0.03 [0.02–0.05]	0.01 [0.01–0.01]	6.7 × 10^−5^	0.02 [0.02–0.04]	0.02 [0.02–0.02]	>0.05	0.03 [0.02–0.04]	0.02 [0.02–0.02]	>0.05
As	0.01 [0.01–0.02]	0.01 [0.00–0.01]	>0.001	0.01 [0.01–0.01]	0.00 [0.00–0.00]	6.0 × 10^−6^	0.01 [0.01–0.01]	0.00 [0.00–0.00]	8.0 × 10^−6^
Ni	0.20 [0.06–0.34]	0.04 [0.01–0.04]	5.8 × 10^−4^	0.05 [0.02–0.07]	0.03 [0.01–0.03]	>0.01	0.04 [0.02–0.10]	0.04 [0.02–0.04]	>0.05
Cd	0.26 [0.03–1.34]	0.03 [0.02–0.03]	>0.01	0.00 [0.00–0.00]	0.00 [0.00–0.00]	>0.05	0.06 [0.00–0.10]	0.00 [0.00–0.00]	9.7 × 10^−4^
Hg	0.01 [0.00–0.01]	0.00 [0.00–0.00]	>0.05	0.01 [0.00–0.02]	0.00 [0.00–0.00]	>0.01	0.00 [0.00–0.03]	0.01 [0.01–0.01]	>0.05

Note: data are presented as median value [Q1–Q3]. *p* values between groups were calculated using the Mann–Whitney U test.

**Table 5 biology-13-00975-t005:** Associations between element content in chicken samples * and bacterial phyla abundance in chicken’s cecal microbiomes, assessed by the coefficient of determination R^2^ **.

Element	Bacterial Phyla in Chicken’s Cecal Microbiomes ***									
	A	B1	B2	C1	C2	M	P	T	V	Unc
Macro elements										
Na		0.43(L)/0.46(B)	0.43(L)/0.46(B)							
K		0.47(L)/0.47(T)/0.47(B)	0.47(L)/0.44(T)/0.45(B)		0.39(B)					
Mg										0.40(L)/0.39(T)
Ca		0.41(L)	0.40(L)	0.52(B)						0.43(T)
P		0.41(T)/0.39(B)	0.40(T)/0.39(B)							0.50(L)
Essential trace elements										
Li							0.70(L)/0.42(T)/0.76(B)			
Sr										
Si					0.42(L)	0.44(L)				
Se										0.46(L)
I		0.51(L)	0.52(L)							0.54(B)
V	0.41(L)	0.64(L)/0.40(T)	0.69(L)							
Cr		0.74(T)	0.72(T)			0.46(L)				
Mn		0.39(T)/0.43(B)	0.42(B)							0.52(L)
Fe	0.43(B)							0.41(B)		0.48(L)
Co										
Cu				0.57(T)						
Zn										0.48(T)/0.40(B)
Toxic elements										
B										0.52(L)
Al				0.71(T)/0.82(B)						
Sn				0.80(L)						
Pb		0.43(L)	0.42(L)							
As		0.55(T)/0.71(B)	0.53(T)/0.66(B)		0.44(T)/0.40(B)		0.39(L)			
Ni								0.39(T)		0.40(L)/0.59(T)
Cd				1.00(T)						0.57(L)/0.52(B)
Hg							0.40(T)			

Note: * Significance (*p* < 0.001) for element content in liver (L), thigh (T) and breast (B). ** Black—positive influence. Red—negative influence. Empty cells (gray area)—no significant influence. *** Column designations: A—*Actinomycetota*; B1—*Bacillota*; B2—*Bacteroidota*; C1—*Chloroflexota*; C2—*Cyanobacteriota*; M—*Mycoplasmatota*; P—*Pseudomonadota*; T—*Thermodesulfobacteriota*; V—*Verrucomicrobiota*; Unc—*unclassified Bacteria*.

## Data Availability

The raw sequencing data may be obtained upon request by e-mail to icis-ofrc@list.ru, belonging to the Institute for Cellular and Intracellular Symbiosis (ICIS) of the Ural Branch of the Russian Academy of Science (Orenburg, Russia).

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
