# Peer review of "Macro, Trace and Toxic Element Composition in Liver and Meat of Broiler Chicken Associated with Cecal Microbiome Community"

_biology, 2024, doi:10.3390/biology13120975_

Round 1

Reviewer 1 Report

Comments and Suggestions for Authors

-Please find my comments to the Ms biology-3316872 entitled "Macro, Trace and Toxic Elements Composition in Liver and Meat of Broiler Chicken Associated with Cecal Microbiome Community”

The Ms addressed here presents a meta-analysis of the detailed relationship between the 25 essential and toxic element compositions in chicken tissues (liver, thigh meat, and breast meat) examined by ICP-MS and the gut microbial community analyzed using NGS techniques.

-The research question is important to determine the relationship between minerals (macro-minerals), essential trace minerals, and toxic elements for a deep understanding of the nature of the relationship (+, - or not exist). This is an invitation area of research where further understanding of this research is needed for both mineral nutrition and the gut ecosystem. 

This article is an invitation field of poultry research that will provide a novel strategy and fill the gap between mineral nutrition and gut microbiota to improve deficiency or excess of mineral macro elements (Na, K, Mg, Ca, and P), trace elements (Sr, Se, Mn, Fe, Co, Cu, Zn), and some toxic elements (B, Pb, Ni, Cd) in the host by gut microbiome modulation, which needs to be confirmed with further in vivo research.

- In the methodology section, these improvements should be considered

-The sex of the animal should add

As mentioned above, the details of the photoperiod program should be described.

-In the methodology section, L 119-120, the form of the diet should be added.

- In addition, the authors should indicate that the control diet is free of antibiotics or phytogenic to avoid the interaction between the gut microbiota and nutrition.

- In the methodology section, L 118, the WHICH nutritional recommendation table was followed?

For the statistical analyses section, the authors should add more details to the experimental model, experimental unit, and power analyses.

• The conclusion section reflects the research outcomes and suggests areas for further research.

• The authors provided an adequate number of most recent (2020-2024) references that fit the Ms content; however, add more 2024 references if they exist.

•  For Table 1, I suggest that the authors add an additional column for the mineral contents of feed samples and, if possible, that of excreta.

 There are minor comments:

1.       L 111, indicate sex of animal

2.       L 122, Plz declare if the basal diet contained any antibiotic or AGP or phytogenic additives

3. L 230, change (Korish et al.,  2020) as (Korish and Attia, 2020)

Reviewer 2 Report

Comments and Suggestions for Authors

 I reviewed the manuscript titled "Macro, Trace and Toxic Elements Composition in Liver and Meat of Broiler Chicken Associated with Cecal Microbiome Community" and summarized my comments as follows:

Lines 106-110: The authors used only two chicken crosses (Arbor Acres and Smena-8), limiting the study's genetic variability. The results could have benefited from using a diverse group of strains (Ross, Cobb, etc.) to account for the diverse broiler strains available in the market.

Line 112: The sample size of 15-30 birds used in the experiments is small. My concern is not getting enough power to detect differences. Did the authors calculate the required sample size before conducting the experiments?

Line 113: The authors stated that all birds were kept in specialized cages with a mesh floor. In the industry, broilers are usually kept on a concrete floor covered with pine shavings. The type of housing system could affect the cecal microbiota and mimicking the industry in such practices could improve the practical significance of the study.

Lines 130-132: no details were given regarding the site from which the samples were collected from each organ. Variations here might impact elemental composition analysis. It is important to clarify this point.

Reviewer 3 Report

Comments and Suggestions for Authors

Dear 

Thank you for allowing me to review the manuscript entitled "Macro, Trace and Toxic Elements Composition in Liver and Meat of Broiler Chicken Associated with Cecal Microbiome Community”.

This study aimed to perform a meta93 analysis of summarized data to explore the detailed relationship between 25 essential and 94 toxic elements composition in chicken’s tissues and observed gut microbial community.

The article is well written and contributes significantly to the field of animal toxicology of chickens.

The article presents a total of 8 figures and 5 tables, some of which are even too long. I also noticed that the discussion is weak and needs to be revised. I ask the authors to explore their data better and discuss the data obtained more clearly.

Below I describe some suggestions.

Sugestion in title: Toxic Elements Composition in Broiler Chicken Associated with Cecal Microbiome Community

Line 49-50: For these reasons, broiler 50 chickens are currently one of the main sources of animal protein [2] - Include citation below

SANTOS, A. N. A. ; VIANA, A. L. ; GUIMARAES, C. C. ; GOMES, M. F. S. ; BARAI, A. A. ; NOBREGA, T. C. ; RIBEIRO, M. W. S. ; SILVA, A. J. I. ; CHAVES, F. A. L. ; MENDONCA, M. A. F. ; SILVA JUNIOR, J. L. ; COSTA NETO, P. Q. ; RUFINO, J. P. F. ; OLIVEIRA, A.T. Paullinia cupana peel meal on the growth performance, meat quality, and haematological and serum biochemical parameters of slow-growing broilers. Animal Production Science, v. 64, p. 1-11, 2024.

Line 55: (...) zinc (Zn) and etc. [4] replace with (...) and zinc (Zn) [4]

Line 58: (As), cadmium (Cd), etc., (...) replace with (As) na cádmium (Cd) (...)

Line 100-104: What is the process number?

Line 112: “Birds” replace with “poultries”

Line 113: “4050 cm2” replace with “4050 cm2

Line 210-216 - A total of 36 broiler chicken’s groups (7 control and 29 experimental), representing 9 experimental series, were included in the study. A total of 25 elements, including 5 macro elements (Na, K, Mg, Ca, P), 12 essential trace elements (Li, Sr, Si, Se, I, V, Cr, Mn, Fe, Co, 213 Cu, Zn) and 8 toxic elements (B, Al, Sn, Pb, As, Ni, Cd, Hg) were determined in liver, thigh and breast samples. Complete data sets were collected for 29 broiler chicken groups, while  data for 7 groups were incomplete for Sr, Al, Sn, Pb, Cd and Hg, among which As was not  determined in 4 groups.

This is methodology

Table 1, line 1: “Maximum permissible level” replace with “Maximum permissible level (FAO/WHO)”

Table 1: If possible, I suggest changing the values ​​to mg Kg-1. Standardize decimal places. Please include standard deviation.

Figure 4: Please delete the outliers. I see them clearly in A, P, T and V.

Table 2: In abundance please include the standard deviation.

Reviewer 4 Report

Comments and Suggestions for Authors

This paper accurately investigated the relationship between 25 macro-, trace- and toxic elements in chicken liver and meat (thigh and breast) and the gur microbial community.

The study identified 3 typical elemental compositions, called α, β and γ, and 2 enterotypes based on the different microbes that were identified. The research results proved the association between the gut microbiome and the elemental composition of the analysed broilers’ tissues proposing a new strategy to contrast the deficiency or excess of certain elements modulating the gut microbial population.

The main topic of this article is interesting and innovative, however there are some small issues to assess before publication:  

-        Line 128: please replace “…the current content is…” with “…the current content to specify…”

-        Line 244: please replace “inter-element’s tendency” with “inter-elements’ tendency”

-        Line 567: Please replace “Comparison of…” with “Comparing…”

Round 2

Reviewer 3 Report

Comments and Suggestions for Authors

The manuscript is ready to be published. Congratulations to the authors for this important scientific contribution.